# Biomimetic Rose Petal Structures Obtained Using UV-Nanoimprint Lithography

**DOI:** 10.3390/polym14163303

**Published:** 2022-08-13

**Authors:** Sruthi Venugopal Oopath, Avinash Baji, Mojtaba Abtahi

**Affiliations:** 1Department of Engineering, La Trobe University, Bundoora, VIC 3086, Australia; 2School of Engineering, Macquarie University, Macquarie Park, NSW 2113, Australia

**Keywords:** superhydrophobic, bioinspired, biomimetic, UV-nanoimprint lithography

## Abstract

This study aims to produce a hydrophobic polymer film by mimicking the hierarchical micro/nanostructures found on the surface of rose petals. A simple and two-step UV-based nanoimprint lithography was used to copy rose petal structures on the surface of a polyurethane acrylate (PUA) film. In the first step, the rose petal was used as a template, and its negative replica was fabricated on a commercial UV-curable polymer film. Following this, the negative replica was used as a stamp to produce rose petal mimetic structures on UV curable PUA film. The presence of these structures on PUA influenced the wettability behavior of PUA. Introducing the rose petal mimetic structures led the inherently hydrophilic material to display highly hydrophobic behavior. The neat PUA film showed a contact angle of 65°, while the PUA film with rose petal mimetic structures showed a contact angle of 138°. Similar to natural materials, PUA with rose petal mimetic structures also displayed the water pinning effect. The water droplet was shown to have adhered to the surface of PUA even when the surface was turned upside down.

## 1. Introduction

Millions of years of evolution have perfected the surface topography of natural materials such that they display unique wettability behavior [1,2,3,4]. For instance, the surface of a lotus leaf has evolved to repel water and dirt [5,6]. Its surface is a combination of micro- and nanoscale hierarchical structures that are made up of low surface energy materials [6]. The surface of a lotus leaf resembles hills and valleys and is responsible for the mechanism that produces the superhydrophobic effect. The contact angles of water droplets on a lotus leaf are reported to be greater than 150° [7,8]. When a drop of water meets the surface, it only wets the air molecules trapped between the valleys and the tip of the hills. These surface properties result in a low coefficient of friction and low adhesive force, making it water-repellent [5]. Additionally, when the lotus leaf is tilted, the water droplet rolls off from the surface, removing dust and pollutants from its surface. In another example, rose petals have attained superhydrophobic behavior due to the presence of hierarchical structures on their surface [9,10,11]. The surface of rose petals consists of micron-size structures called papillae, with nanofoldings called cuticles present on each of the papillae [9]. Although the water droplets do not spread on the surface of rose petals, water droplets stay adhered to the surface of the rose petals, even when the surface is turned upside down. This ‘petal effect’ is attributed to the topography present on the rose petals. Water droplets stay adhered to the surface as the surface displays the ‘Cassie impregnating’ state [11]. The water enters the region between the micropapillae, and it stays adhered to the surface due to the large contact area. Despite this, the material displays superhydrophobicity due to the presence of air in between the nanofoldings on top of each papilla.

Many studies have mimicked the surface topography found on natural materials to produce bioinspired, highly hydrophobic, and superhydrophobic materials [1,2,12]. These materials have found applications in a wide variety of areas including self-cleaning surfaces, oil–water separation, drag reduction, and anti-biofouling surfaces [2,3,10]. Typically, studies have incorporated the use of both chemical and topographical modifications to fabricate highly hydrophobic and superhydrophobic surfaces [13,14,15,16]. For example, some studies have first fabricated micro- and nanoscale structures on the surface of the polymer and then altered the surface energy by treating the surface with either some low-energy chemicals or plasma for it to display highly hydrophobic or superhydrophobic behavior [13,15,17,18]. A wide variety of fabrication techniques including hot embossing, laser-based structuring, and photolithography have been adopted to introduce micro/nanoscale structures on the surface of polymer films to display superhydrophobicity [18,19,20,21,22,23]. Among these, hot-embossing and nanoimprint lithography are the methods most commonly used for the fabrication of reproducible and high-resolution micro- and nanoscale hierarchical structures. Many of these studies have replicated the hierarchical micro/nanostructures found on natural materials such as butterfly wings, gecko feet, rose petals, lotus leaf, etc. onto thin polymer films to produce superhydrophobic materials [24,25,26].

In this article, we used a UV-based nanoimprint lithography technique to fabricate a highly hydrophobic biomimetic surface. The hierarchical micro/nanostructures on the polymer film were produced by copying the microstructure found on the surface of rose petals. Our results demonstrate that introducing rose petal mimetic microstructures on a surface of an inherently hydrophilic material can help them display highly hydrophobic surface behavior. These surfaces are shown to display highly hydrophobic behavior without any chemical treatment of the surface. Additionally, the study demonstrates that coating the surface with a monolayer of a low surface energy silane can enable the material to display superhydrophobic behavior.

## 2. Materials and Methods

### 2.1. Materials

A commercially available UV-curable epoxy, Norland Optical Adhesive (NOA-61) was obtained from AusOptic (Macquarie Park, NSW, Australia) and was used to obtain the negative rose petal template. Positive rose petal structures were produced on photo-curable polyurethane-acrylate (PUA) precursor (MINS-301RM) obtained from Minuta Tech. Co. Ltd. (Songnam, Korea).

### 2.2. Inverse Rose Petal Structures

A fresh rose petal obtained from Victoria, Australia was affixed onto a glass substrate and was used as a master template. No anti-adhesive coating was applied on the rose petal as coating the surface of the rose petal with a monolayer of silane collapsed the surface structures. Following this, UV-curable NOA61 was carefully spread onto the surface of the rose petal such that a uniform layer was obtained. A thin optically translucent Mylar polyester film was placed on top of the UV-curable NOA61 layer. To produce the inverse rose petal structures on NOA61, the setup was placed in a Compact Imprinting tool (CNI, NIL Technology ApS, Kongens Lyngby, Denmark) to cure NOA61 with UV light. Using the CNI tool, the setup was exposed to a UV source for 1 min (50% power). A vacuum pressure of 1 bar was applied before the setup was exposed to the UV source. Once the NOA61 layer was fully cured, NOA61 with the Mylar substrate was easily peeled off from the surface of the rose petals. The schematic shown in Figure 1 displays the steps followed to produce the inverse rose petal structures. 

### 2.3. Rose Petal Structures

NOA61 with inverse structures was then used as a stamp to produce positive rose petal structures. A UV-curable polymer (PUA) was evenly spread onto the inverse of the rose petal structures produced on the NOA61 template, and an optically translucent Mylar film was placed on top of the PUA layer, as shown in Figure 1. To ensure that cured PUA can be easily detached from the NOA61 template, the edges of the NOA61 template were covered with Teflon tape. The CNI tool was used to imprint the structures onto the PUA film by exposing the setup to a UV source. A vacuum pressure of 2 bar was applied before UV exposure to allow the PUA to flow into the cavities present on the NOA61 template. Once the PUA film was fully cured, the Teflon tape that was fixed around the edges of the template was peeled off carefully, which also helped to detach the cured PUA film from the NOA61 template. The cured PUA film revealed the replica of rose petal structures on the surface of the PUA film. 

### 2.4. Characterization

The microstructure of the samples was examined using an ultra-high resolution Schottky scanning electron microscopy (FESEM SU7000, Hitachi, Tokyo, Japan) at an accelerating voltage of 10 kV. The samples were sputter-coated with a thin layer of gold (18 mA, 60 s) before they were examined using SEM.

Contact angle measurements on samples were made by placing a ~3 µL water droplet on the surface of the sample and by measuring the angle made by the droplet. 

## 3. Results and Discussion

The surface of the rose petal comprises hierarchical structures that give them a unique wettability behavior [11]. The surface of the rose petal consists of a tightly packed array of micrometer-sized bumps, called papillae [27,28,29]. Nanometer-sized wrinkles, called cuticles, are present on top of each of these papillae. The diameter of papillae is found to range between 20 and 30 µm [27]. The presence of these hierarchical micro/nanostructures on the rose petals helps them to display a water contact angle of ~152°. What is more interesting is that the droplets of water cling onto the surface of the rose petals and remain immobile even when the surface is inverted. The wettability behavior of the rose petal’s surface can be reproduced on polymer films and surfaces by mimicking the hierarchical micropapillae and nano-cuticle microstructure. 

UV-based nanoimprint lithography is used to copy the rose petal’s microstructure onto a polymeric thin film. The fabricated negative replica on an NOA61 film is examined using scanning electron microscopy (SEM). Figure 2a shows the inverse of rose petal structures produced on NOA61 film, which is obtained using the first nanoimprint lithography step. A SEM image shows that the structures found on the surface of the NOA61 film are similar to the tightly packed structure array that is commonly found on the surface of rose petals. The micrometer-sized concave structures evident in the SEM image are the negative replica of the micropapillae found on the surface of rose petals. The average diameter of each honeycomb-like structure is determined to be ~24 µm using the ImageJ software. The higher magnification image shows the presence of nanometer-sized cavities within each of these concave structures. These nanometer-sized cavities are the negative replica of the nanofolds found on the surface of each of the micropapillae. Following this, the fabricated NOA61 with a negative replica of rose petal structures is taken as a template and its microstructure is transferred to the surface of a PUA film. This step led to producing a positive replica of rose petal structures on the surface of PUA. Figure 2b shows the positive replica of rose petal structures obtained on the PUA film. 

It is evident from Figure 2b that the microstructure formed on the PUA film is similar to the kind of microstructure found on the surface of rose petals. Figure 2b demonstrates that the positive structures formed on PUA well copied the rose petal’s surface topology. A very good pattern transfer uniformity is observed in both samples (negative and positive replicas). The width of these structures measured using ImageJ software is determined to range between 16.5 and 23.6 µm. This shows that the size of the structure has slightly decreased, which can be attributed to the shrinkage of PUA during the curing stage. The higher magnification image demonstrates the presence of dense nanofoldings on each of the papillae. The width of these nanofoldings is determined to vary from 300 to 500 nm. These results are in agreement with the results reported in the literature on rose petal structures. A similar lithography technique is used by Wang et al. [11] to copy the rose petal structures onto a polymer film. The hierarchical microstructure of rose petals is copied on PDMS using a two-step casting technique. Briefly, an aqueous solution of polyvinyl alcohol (PVA) is poured on a fresh rose petal. The PVA film with negative rose petal structures is obtained after the water is completely evaporated. Following this, PDMS is poured onto the negative replica of the rose petal structures formed on PVA. After curing, the PDMS contained the positive rose petal structures consisting of micropapillae with nanofolds.

In the next step, we investigated the role of rose petal mimetic structures on the surface wettability of a PUA film. As a control sample, we first fabricated a PUA film that had no surface structures and investigated its wettability behavior. Figure 3a shows that the contact angle measured on a neat PUA film is determined to be ~65°, indicating that the material is hydrophilic in nature. A similar contact angle value of ~65 ± 5° is reported in the literature on neat PUA [30]. In order to determine the role of microstructures on the wettability behavior of PUA, we also measured the contact angle on PUA that had a negative replica of rose petal structures. To prepare these samples, PUA prepolymer is placed on the surface of fresh rose petals followed by UV-nanoimprinting. This resulted in the formation of inverse rose petal structures on PUA. Figure 3b shows the contact angle measured on this sample. Interestingly, the contact angle on the inverse rose petal surface formed on PUA is determined to be ~115°. Typically, introducing roughness on an inherently hydrophobic material improves the hydrophobicity of the material. Similarly, introducing roughness to an inherently hydrophilic material should improve its hydrophilicity. Our results on the contrary show that the inherently hydrophilic PUA is made hydrophobic by introducing inverse rose petal structures on its surface. The inverse rose petal structures formed on PUA had microcavities on their surface, which traps air molecules. Such surfaces display hydrophobic behavior as the presence of air enlarges the water/air interface and minimizes the solid/water interface. Hence, the water droplet sits on the air molecules trapped in the cavities of the samples. The air molecules inhibit the water from wetting the surface. Similar results have been reported in the literature where they show that introducing micro/nano textures on inherently hydrophilic materials can enable them to display hydrophobic behavior. For example, Ems et al. [31] show that the hydrophilic silicon substrate can be made hydrophobic by introducing cavities onto the surface of the silicon substrate. In another study, Zhu et al. [12] show that the presence of microscale hill and valley structures on the surface of an inherently hydrophilic film enables it to display superhydrophobic nature. Hou et al. [32] demonstrated that hydrophilic PUA can be made to display hydrophobic behavior by introducing surface structures to it. They also coated the PUA with siloxane and demonstrated that PUA can display superhydrophobicity. They argue that the wetting behavior of this sample can be attributed to the Cassie–Baxter state. When the distance between the micropillars increased beyond 70 µm, the wetting state was noticed to change from the Cassie–Baxter state to the Wenzel state. 

The fabricated PUA film with a negative replica of rose petals displays the Cassie–Baxter wetting state due to the proximity of the cavities. The water droplet is restricted from entering the cavities and hence the sample displays hydrophobic nature. Following this, the contact angle is measured on PUA film that had a positive replica of rose petal structures. Positive rose petal structures on PUA are prepared as described in Figure 1. The contact angle on this sample is measured to be ~138°, implying that the material is highly hydrophobic. The improvement in hydrophobicity can be attributed to the presence of hills and valleys and the presence of nanofoldings on top of micropapillae. The water droplet sits on the air molecules trapped between the nanofoldings present on top of the micropapillae, enabling the sample to display hydrophobicity. This should explain why PUA with a positive replica of rose petal structures displays enhanced hydrophobic behavior compared to neat PUA and PUA with a negative replica of rose petal structures. Similar results are reported in the literature, where an inherently hydrophilic material is shown to display superhydrophobic behavior [33]. It is argued that structured hydrophilic material can display superhydrophobic behavior when water droplet sits at least partially on top of air [33]. Fu et al. [34] show that hydrophobic surfaces can be produced on inherently hydrophilic poly (methyl methacrylate) (PMMA) and polyethylene terephthalate (PET) by introducing micropillar arrays on their surfaces. Their results show the wetting state displayed by these surfaces is between the Wenzel and Cassie–Baxter states. A water droplet on such surfaces rests partly on air pockets present in between the microstructures and partly penetrates the valleys. Thus, the hierarchical structures enhanced the hydrophobicity of the material. 

To further improve the hydrophobicity of our samples, the PUA film with a replica of rose petal structures was coated with a hydrophobic self-assembled monolayer of 1H, 1H, 2H, 2H-perfluorooctyl-trichlorosilane (PTCS). This is achieved by placing a drop of PTCS in a vacuum desiccator along with the sample for ~2 h. The contact angle on this sample is measured to be ~151° (see Figure 3d). The water droplet on this sample is seen to preserve its spherical shape. The superhydrophobic behavior displayed by this sample can be attributed to the low surface energy coating of monolayer silane as well as surface topography.

The ‘petal effect’ of the fabricated sample is also investigated. The ability of the rose petal mimetic PUA surface to pin the water droplets on the surface is demonstrated in Figure 4. The water pinning effect is due to the ‘Cassie-impregnating’ state achieved by the water droplet on this surface. In this state, the water droplet penetrates the spaces between the micropapillae but sits on the air trapped between the nanoscale grooves. This helps the water droplet experience a high adhesive force on the sample. Hence, the water droplet appears to stay attached to the sample, even when the surface of the sample is tilted to 90 as well as 180°.

## 4. Conclusions

A double-step replication using UV nanoimprint lithography was used to copy the rose petal structures on the surface of polyurethane acrylate (PUA) films. The wettability behavior of the patterned PUA films was investigated and compared with neat PUA films. The neat PUA film was determined to be hydrophilic. However, its hydrophobicity is shown to increase by introducing both negative and positive rose petal structures on PUA. The increase in hydrophobicity can be attributed to the trapped air molecules between the nanoscale structures. Additionally, introducing rose petal mimetic structures on PUA is shown to result in the ‘Cassie-impregnating’ wetting state. A water droplet placed on this surface is shown to be pinned to the surface and is shown to be immobile even when the surface is turned upside down. 

## Figures and Tables

**Figure 1 polymers-14-03303-f001:**
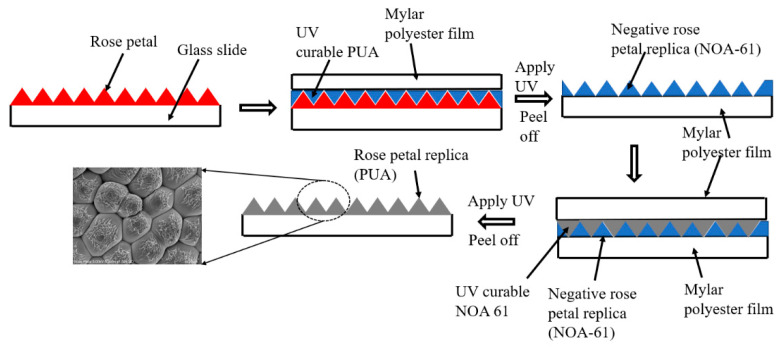
Schematic demonstrating the procedure used to produce rose petal structures on PUA film.

**Figure 2 polymers-14-03303-f002:**
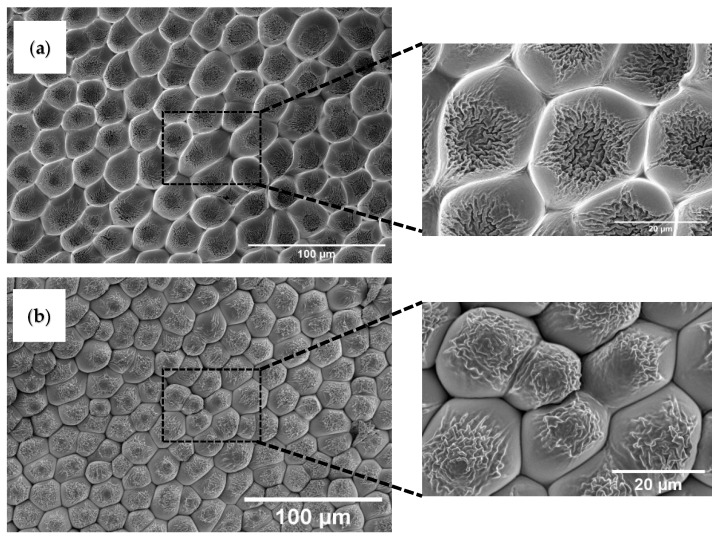
Scanning electron micrography (SEM) images of negative and positive replicas of rose petal structures. (**a**) Image represents the negative replica fabricated on NOA61 film and (**b**) image represents the positive replica fabricated on PUA film.

**Figure 3 polymers-14-03303-f003:**
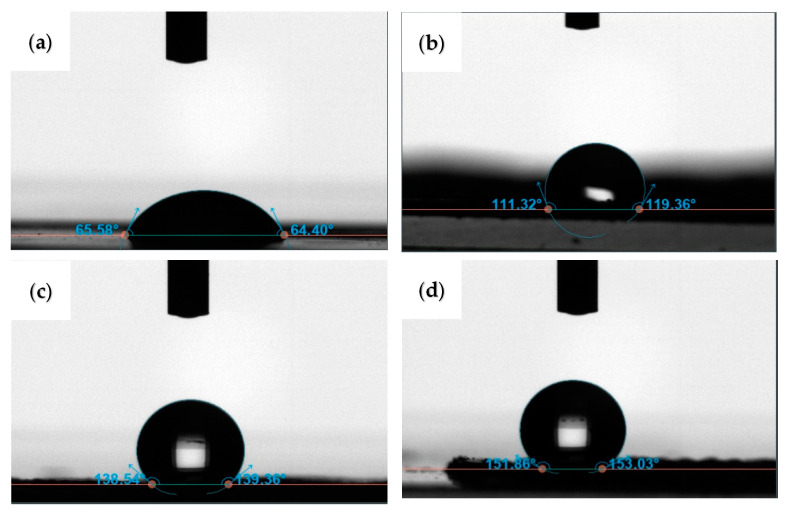
Contact angle measurements made on (**a**) neat PUA film, (**b**) PUA film with negative replica of rose petal structures, (**c**) PUA film with positive replica of rose petal structures, and (**d**) silane treated PUA film with positive replica of rose petal structures.

**Figure 4 polymers-14-03303-f004:**
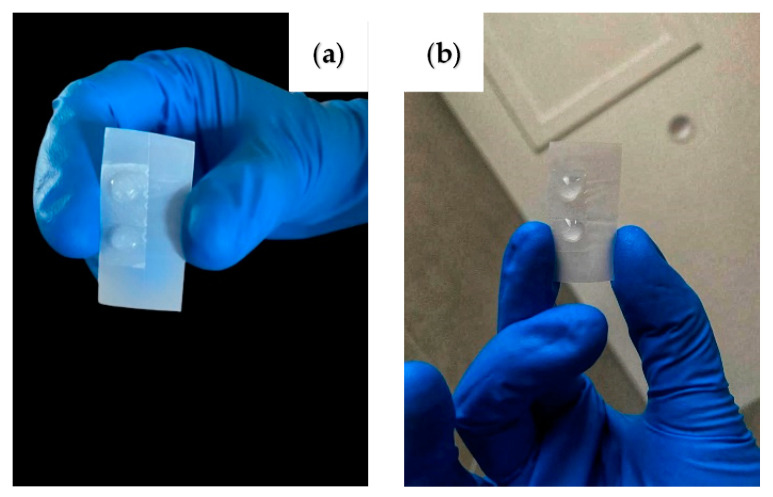
Digital image demonstrating the ‘petal effect’ displayed by the sample when the sample is titled at (**a**) 90° and (**b**) 180°.

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
