# Peer review of "Biomimetic Rose Petal Structures Obtained Using UV-Nanoimprint Lithography"

_polymers, 2022, doi:10.3390/polym14163303_

Round 1
Reviewer 1 Report
You described Biomimetic rose petal structures obtained using UV-nanoim-2 print lithography.
It's many interesting paper, however, you should be described some parameters to publish your paper as follows;
1. To fabricate replica from the master, what kinds of anti-adhesive coating method did you use on the master side and the 1st replica? Also, what kind of adhesive materials did you use between the PUA and the Mylar polyester film?
2. On your experiments, there are no any information for adhesive and anti-adhesive coating materials between two Mylar polyester films in Fingure 1.
3. How much did you get pattern transfer uniformity from the mater or 1st replica?
4. In figure 3, you fabricate the 1st replica with contact angle of 111~119 degree from the master at the 1st step and then fabricate 2nd replica from the 1st repplica with contact angle of 138 degree. Dow did you get this result? Which method did you use to increase the contact angle? As shown in Fig.3(D), if you use the silane coating on the replica, then to get more higher contact angle is too straightforward things.
Author Response
Modifications made in response to comments from Referee 1:
Q1 To fabricate replica from the master, what kinds of anti-adhesive coating method did you use on the master side and the 1st replica? Also, what kind of adhesive materials did you use between the PUA and the Mylar polyester film?
A1 We thank the referee for this suggestion. We have revised the manuscript and explained the steps in the manuscript.
No anti-adhesive coating was used on the rose petal (master stamp). The PUA was found to be adhered to the Mylar film and hence no adhesive was used between PUA and Mylar Film. The role of Mylar film was to provide some stiffness to the sample.
To enable the detachment of PUA from the NOA61 template, Teflon tape was fixed on the edges of the NOA61 template. After PUA was cured, the Teflon tape was slowly peeled off which also helped to detach the PUA from NOA61.
These procedures are explained in the revised manuscript.
Q2 On your experiments, there are no any information for adhesive and anti-adhesive coating materials between two Mylar polyester films in Fingure 1.
A2 We thank the referee for this comment. It is hard to illustrate the procedure adopted in the figure. However, we have explained the procedure adopted in the revised text.
No anti-adhesive coatings are applied at this stage. We did not want to apply any anti-adhesive coating at this stage as we wanted to ensure that the hydrophobicity behavior can only be attributed to the structures that are formed and not be attributed to the presence of low surface energy anti-adhesive layer.
Q3 How much did you get pattern transfer uniformity from the mater or 1st replica?
A3 The authors obtained very good pattern transfer uniformity both from the master and the 1st replica. A random section of the sample was chosen for visualization under SEM and it was observed that the samples had very good pattern transfer uniformity.
Q4 In figure 3, you fabricate the 1st replica with contact angle of 111~119 degree from the master at the 1st step and then fabricate 2nd replica from the 1st repplica with contact angle of 138 degree. Dow did you get this result? Which method did you use to increase the contact angle? As shown in Fig.3(D), if you use the silane coating on the replica, then to get more higher contact angle is too straightforward things.
A3 No Silane coating was applied to get the 2nd replica Figure 3C. We wanted to demonstrate that the wettability of the sample is influenced by the presence of structures and not due to the presence of low surface energy coatings. The increase of contact angle to 138° is attributed to the presence of hills and valleys on the surface of the sample and presence of nanofoldings.
Reviewer 2 Report
The paper entitled "biomimetic rose petal structures obtained using UV-nanoimprint lithography" describe a study to mimic the hierachical micro-/nano-structures to produce hydrophobic substrate with hydrophillic materials. They used UV cured PUA film and double nanoimprints to transfer the hierachical structure from the rose petal to PUA film. The wettability of such structures are studied using created substrate. It was found that using hierachical structures the wettability can be tuned.
The paper is well written scientifically, but require editing. There are many typos and citation style difference. For instance, page 3 section 3 second line has different citation style.
The paper is a duplication of other's past work (e.g. Ref 11, 27, 28 etc.). Scientifically, it does not offer additional insights on wettability of a structured substrate, nor have new mechanisms. Hence, the reviewer is puzzle what is the new claims of this paper to the others.
Technical concern:
NOA61 is a very viscous fluid (>~300 cp). It is difficult to believe that polymer will penetrate into nanopores and local nano-scale features by simply spread or pouring. To fully penetrate the nano-pours, vacuum is often used to remove trapped air. Can author provide evidence that the nanoscale features are adequately produced by comparing petal's SEM and that of PUA replica using their fabrication method?
Author Response
Modifications made in response to comments from Referee 2:
Q1 The paper is well written scientifically, but require editing. There are many typos and citation style difference. For instance, page 3 section 3 second line has different citation style
A1 We thank the referee for pointing this out. The manuscript has been proof-read and the typos, grammatical errors as well as citation style differences have been corrected.
Q2 The paper is a duplication of other's past work (e.g. Ref 11, 27, 28 etc.). Scientifically, it does not offer additional insights on wettability of a structured substrate, nor have new mechanisms. Hence, the reviewer is puzzle what is the new claims of this paper to the others.
A2 This work shows that an inherently hydrophilic material can display hydrophobic nature when its surface is structured, particularly when its surface structures mimic the structures found on rose petals. We believe this aspect of the manuscript is novel compared to the other published studies. In addition, we believe these results are encouraging as they show that potentially superhydrophobicity can be attained on a moderately hydrophobic material by introducing rose petal mimetic structures on its surface.
Q3 NOA61 is a very viscous fluid (>~300 cp). It is difficult to believe that polymer will penetrate into nanopores and local nano-scale features by simply spread or pouring. To fully penetrate the nano-pours, vacuum is often used to remove trapped air. Can author provide evidence that the nanoscale features are adequately produced by comparing petal's SEM and that of PUA replica using their fabrication method?
A3 We agree with this comment of the referee. However, NOA61 is made to penetrate into the pores due to the application of vacuum and pressure during the nanoimprinting process.
Round 2
Reviewer 1 Report
It's well revised according to the comments.